# Targeted scavenging of extracellular ROS relieves suppressive immunogenic cell death

Hongzhang Deng[1,2], Weijing Yang[2], Zijian Zhou[2], Rui Tian[2], Lisen Lin[1], Ying Ma[2], Jibin Song [1✉] & Xiaoyuan Chen [2✉]

Immunogenic cell death (ICD) and tumour-infiltrating T lymphocytes are severely weakened by elevated reactive oxygen species (ROS) in the tumour microenvironment. It is therefore of critical importance to modulate the level of extracellular ROS for the reversal of immuno-suppressive environment. Here, we present a tumour extracellular matrix (ECM) targeting ROS nanoscavenger masked by pH sensitive covalently crosslinked polyethylene glycol. The nanoscavenger anchors on the ECM to sweep away the ROS from tumour microenvironment to relieve the immunosuppressive ICD elicited by specific chemotherapy and prolong the survival of T cells for personalized cancer immunotherapy. In a breast cancer model, elimination of the ROS in tumour microenvironment elicited antitumour immunity and increased infiltration of T lymphocytes, resulting in highly potent antitumour effect. The study highlights a strategy to enhance the efficacy of cancer immunotherapy by scavenging extracellular ROS using advanced nanomaterials.

[1] MOE key laboratory for analytical science of food safety and biology, College of Chemistry, Fuzhou University, Fuzhou 350108, China. [2] Laboratory of Molecular Imaging and Nanomedicine (LOMIN), National Institute of Biomedical Imaging and Bioengineering (NIBIB), National Institutes of Health (NIH), Bethesda, MD 20892, USA. ✉email: jibinsong@fzu.edu.cn; chen9647@gmail.com

Cancer immunotherapies that stimulate the inherent immunological systems of the body to recognize, attack, and eradicate tumour cells have demonstrated varying degrees of success[1–5]. Recent studies revealed that immunogenic cell death (ICD) elicited by specific chemotherapy or radiotherapy makes the dead cell corpses 'visible' to dendritic cells (DCs) that present antigens to T cells with specific antitumour immune responses, which then control residual tumour cells[6–9]. However, ICD induced immune response can be severely weakened and even abolished by elevated reactive oxygen species (ROS) in the tumour microenvironment (TME)[10,11]. In the meantime, T cells become dysfunctional after reaching the tumour site[12–16]. Therefore, how to modulate the level of extracellular ROS is utterly important to reverse the immunosuppressive nature of the TME.

ICD provides dying cancer cells with stimuli to elicit immune responses as a tumour vaccine[17–21]. Under the potent ICD-inducing therapies, damage-associated molecular patterns (DAMPs) are secreted from dying cells or exposed on the outer layer of the cell membrane to facilitate immune responses. Release of adjuvant high mobility group protein B1 (HMGB1) as "find me" signals is essential to induce DC maturation[22–24]. However, HMGB1 is often oxidized by ROS in the TME and its stimulatory activity is thus neutralized[11]. Although there have been strategies reported to enhance tumour cell death in an immunogenic way by increasing the amount of released HMGB1 into the TME, the fate of released HMGB1 has received limited attention[25]. Therefore, how to keep the stimulatory activity of HMGB1 in the TME is a major challenge. Despite the potential, the antitumour immunity triggered by ICD is also limited, which is mainly due to the dysfunctional T cells in the TME. Elevated extracellular ROS controls immune regulation and reduces the proliferation and antitumour function of T cells[26–29]. Therefore, strategies to extend the survival of T cells and recover immune functions by scavenging ROS in the TME are urgently needed.

In this study, we prepared a ROS nanoscavenger (T$^{ECM}$-NS) modified with ECM targeting peptide. Dual-benzaldehyde terminated polyethylene glycol as a caging polymer was introduced to construct the crosslinked "stealth" delivery system (PEG-T$^{ECM}$-NS) with pH sensitive imine bonds. This intelligent nanoscavenger can sweep away the ROS from the TME to relieve the immunosuppressive ICD elicited by oleandrin (OLE) anticancer drug and prolong the survival of T cells for "personalized" cancer immunotherapy (Fig. 1). When arriving at the tumour site, the de-shielding of PEG corona triggered by tumour acidity leads to the exposure of ECM targeting peptide and anchoring on the ECM, allowing T$^{ECM}$-NS to continuously scavenge extracellular ROS. Meanwhile, T$^{ECM}$-NS is oxidized by ROS, inducing the disruption of nanoparticles to release OLE. Free OLE is internalized into tumour cells and induces ICD through the release of HMGB1 from the dying cells. More importantly, due to the extracellular ROS neutralization by nanoscavenger, HMGB1 can maintain its stimulatory activity to realize DC activation by dying or dead cells and antigen processing to T cells with effective antitumour immune response.

## Results

### Preparation and characterization of ROS nanoscavenger.
Poly(propylene sulfide) (PPS) was synthesized via anionic polymerization of propylene sulfide using 3-mercaptopropionic acid as an initiator (Supplementary Fig. 1). The chemical structure and composition of PPS were measured by $^1$H NMR, $^{13}$C NMR, and gel permeation chromatography (GPC) (Supplementary Figs. 2–4). Polyetherimide-g-poly(propylene sulfide) (PEI-PPS) was synthesized via the amidation reaction of PEI and PPS in the presence of (2-(1H-benzotriazol-1-yl)-1,1,3,3-

tetramethyluronium hexafluorophosphate/N,N-diisopropylethylamine (Supplementary Fig. 5). The chemical structure of PEI-PPS was confirmed by $^1$H NMR (Supplementary Fig. 6). PEG with dual aldehyde end groups was obtained by the esterification reaction of PEG and p-formylbenzoic acid in the presence of dicyclohexylcarbodiimide/4-dimethylaminopyridine (Supplementary Fig. 7). The structure and composition were characterized by $^1$H NMR, $^{13}$C NMR, and GPC (Supplementary Figs. 8–10). Collagen targeting peptide (T$^{ECM}$) was prepared by automated solid-phase peptide synthesis at a 100 μmol scale on a rink amide resin using standard Fmoc-based protocols[30] (Supplementary Fig. 11). The structure of peptide was confirmed by high performance liquid chromatography (HPLC) and liquid chromatography–mass spectrometry (Supplementary Figs. 12 and 13). Nanoparticle of PEI-PPS was prepared by a dialysis method, termed as nanoscavenger (NS). Then, tumour collagen targeting peptide (T$^{ECM}$) was conjugated onto the surface of PEI-PPS nanoparticles through an amidation reaction, termed as T$^{ECM}$-NS (Fig. 2a). The conjugation efficiency is about 90%. The "stealth" nanoparticles (PEG-T$^{ECM}$-NS) were prepared and crosslinked by the PEGylation of CHO-PEG-CHO via pH sensitive benzoic imine bond on the surface of T$^{ECM}$-NS. The coating efficiency of PEG on the surface of nanoparticles was about 100% (Supplementary Fig. 14). The morphology of NS, T$^{ECM}$-NS, and PEG-T$^{ECM}$-NS were examined using transmission electron microscope (TEM) (Fig. 2b). TEM result indicated that NS, T$^{ECM}$-NS, and PEG-T$^{ECM}$-NS have spherical structures and the average diameters were about 100 nm. The size distribution and zeta potential of NS and T$^{ECM}$-NS were evaluated by a laser particle size analyzer (Fig. 2c, d). As shown in Fig. 2e, the zie of nanoparticles remained unchanged after coating PEG on the surface of T$^{ECM}$-NS. However, the zeta potential was decreased with increasing amount of PEG (Fig. 2f). Because the binary PEG-T$^{ECM}$-NS with T$^{ECM}$-NS/PEG mass ratio of 10:2 could efficiently shield the positive charge of T$^{ECM}$-NS, we chose this particular formula for further experiments. Bovine serum albumin (BSA) incubation (Supplementary Fig. 15) showed no obvious adsorption of BSA, indicating the potential high stability of PEG-T$^{ECM}$-NS in blood circulation, due to the protection by PEG caging polymer.

Then, the ROS scavenging capacity and collagen binding ability of PEG-T$^{ECM}$-NS were evaluated. The characteristics of OLE loaded PEG-T$^{ECM}$-NS were summarized (Supplementary Table S1). The accumulated drug release curve of PEG-T$^{ECM}$-NS at pH 7.4 and 6.8 with or without 10 mM H$_2$O$_2$ was shown in Fig. 2g. The drug release of PEG-T$^{ECM}$-NS at pH 7.4 with or without 10 mM H$_2$O$_2$ was negligible and less than 25% in 24 h, indicating that the crosslinked structure of PEG-T$^{ECM}$-NS was relatively stable under neutral condition. The accumulative amount of OLE from PEG-T$^{ECM}$-NS at pH 6.8 in the presence of 10 mM H$_2$O$_2$ was 3.21-fold higher than that at pH 7.4 with or without 10 mM H$_2$O$_2$ in 24 h. The release rate of OLE was significantly enhanced at pH 6.8 with the presence of 10 mM H$_2$O$_2$, under which the PEG coating on the surface of T$^{ECM}$-NS was de-shielded due to the breakage of pH sensitive imine bonds. Despite the removal of PEG, T$^{ECM}$-NS was able to maintain the intact nanoparticle structure (Supplementary Figs. 16 and 17). Therefore, without H$_2$O$_2$, the drug release rate of PEG-T$^{ECM}$-NS at pH 6.8 was only about 28% and slightly faster that under physiological condition (about 23%). The significantly increased OLE release from PEG-T$^{ECM}$-NS at pH 6.8 with 10 mM H$_2$O$_2$ (Fig. 2g) was attributed to pH-induced de-shielding of PEG and hydrophilic transition of thioether to sulfoxide, inducing the disassembly of the nanoparticles. Through the oxidation of thioether residues in PPS segments to sulfoxides, PEG-T$^{ECM}$-NS have the capacity to scavenge ROS (Supplementary Fig. 18). To assess the ROS

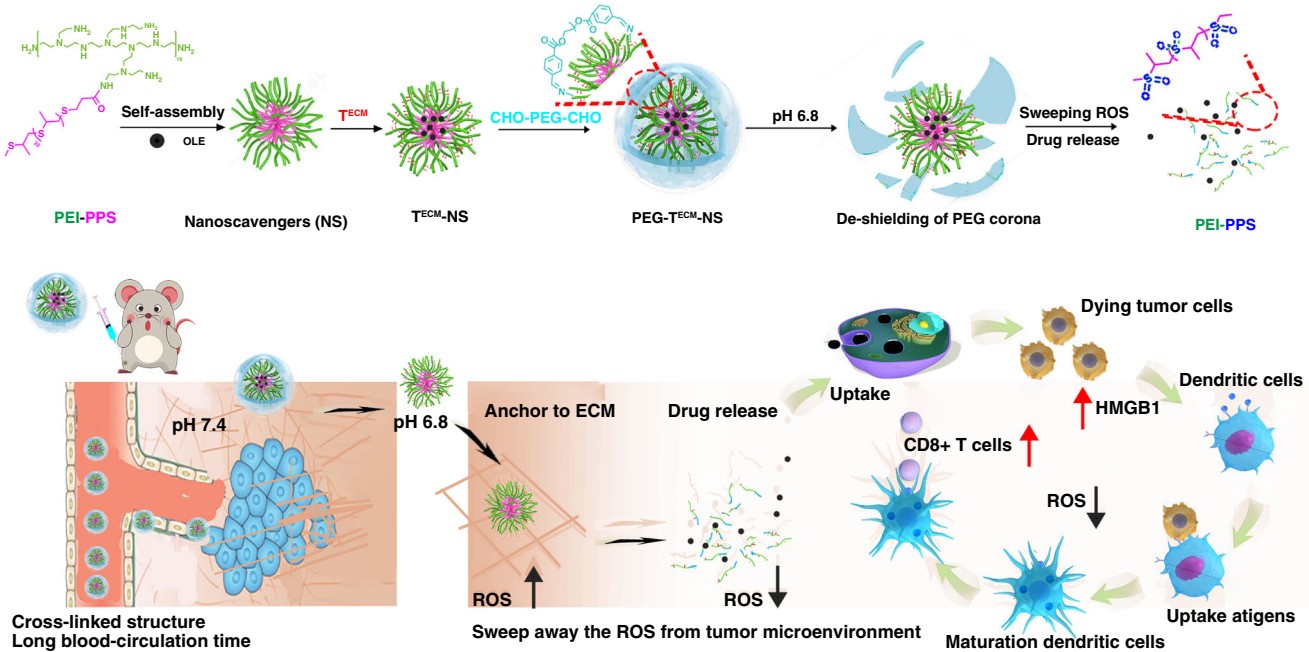

**Fig. 1 Extracellular ROS nanoscavenger reverses immunosuppression.** Schematic illustration of tumour extracellular matrix (ECM) targeted ROS nanoscavenger masked by pH sensitive covalently crosslinked polyethylene glycol. The nanoscavenger anchored on the ECM sweeps away the ROS from tumour microenvironment to relieve the immunosuppressive ICD elicited by specific chemotherapy and prolong the survival of T cells for personalized cancer immunotherapy.

scavenging capability of PEG-$T^{ECM}$-NS, the residual $H_2O_2$ concentration after incubation with PEG-$T^{ECM}$-NS was detected using a $H_2O_2$ assay kit. Fluorescence at 562 nm was measured as a function of $H_2O_2$ concentration in the presence of PEG-$T^{ECM}$-NS pretreated at pH 6.8. There is a sharp decrease in fluorescence at 562 nm in the presence of nanoparticles, suggesting strong $H_2O_2$ scavenging activity of PEG-$T^{ECM}$-NS and $T^{ECM}$-NS (Fig. 2h and Supplementary Fig. 19). As a control, we prepared PEI grafted with $C_{18}$, which was conjugated with $T^{ECM}$ on the surface of PEI-$C_{18}$ nanoparticles ($T^{ECM}$-$C_{18}$). PEG was also coated on the surface of $T^{ECM}$-C18 (PEG-$T^{ECM}$-$C_{18}$). Parallel studies showed that PEG-$T^{ECM}$-$C_{18}$ was incapable of scavenging ROS (Supplementary Fig. 20).

Time-dependent de-shielding of PEG from PEG-$T^{ECM}$-NS was evaluated by $^1$H NMR. At pH 6.8, after different incubation times, PEG-$T^{ECM}$-NS was centrifuged in a centrifugal dialysis tube (molecular weight cut-off: 30,000 Da). The supernatant was freeze dried and the PEG content was measured by $^1$H NMR (Fig. 2i). Due to the relatively rapid hydrolysis rate of the imine linker between PEG and $T^{ECM}$-NS, about 85% of PEG was removed from PEG-$T^{ECM}$-NS within 2 h. The binding affinity of PEG-$T^{ECM}$-NS and $T^{ECM}$-NS to collagen was measured by an enzyme-linked immunosorbent assay (ELISA) using biotinylated nanoparticles and substrate coated with rat tail collagen type I (Fig. 2j). $T^{ECM}$-NS with collagen targeting peptides on the surface showed much higher binding affinity with collagen than PEG-$T^{ECM}$-NS. When PEG-$T^{ECM}$-NS was pretreated with pH 6.8 buffer for 2 h to de-shield PEG, the collagen binding affinity was recovered to a similar level with $T^{ECM}$-NS. These results demonstrated that PEGylated PEG-$T^{ECM}$-NS can protect the collagen targeting peptides and improve targeting selectivity. It is expected that the crosslinked PEG on the surface of $T^{ECM}$-NS can shield the collagen targeting peptides during blood circulation. Upon arrival at the tumour site, the rapid de-shielding of PEG corona from the nanoparticles triggered by the TME will lead to the exposure of the collagen targeting peptides to

anchor the nanoparticles onto the ECM. Isothermal titration calorimetry (ITC) is a sensitive technique that measures the heat of reaction of two aqueous solutions when one is titrated against the other[31–35]. As shown in Fig. 2k, each peak represents the enthalpy change caused by the peptide–collagen interactions. The exothermicity of the interaction of the peptide during titration with collagen is very high, indicating the strong binding ability. Furthermore, we also evaluated the selective collagen binding properties of PEG-$T^{ECM}$-NS and $T^{ECM}$-NS in the heterogeneous environment of 4T1 murine mammary tumour. After 4T1 tumour pieces (1 cm$^3$) were incubated with $T^{ECM}$-NS and PEG-$T^{ECM}$-NS at pH 7.4 and pH 6.8, respectively, confocal laser scanning microscopy (CLSM) images of the sectioned slices from the 4T1 tumour were examined. The complete overlap of $T^{ECM}$-NS or PEG-$T^{ECM}$-NS signal at pH 6.8 with anti-collagen antibody further confirmed the collagen specificity of $T^{ECM}$-NS (Fig. 2l and Supplementary Fig. 21).

**Investigation of PEG-$T^{ECM}$-NS/OLE-induced ICD and in vitro anticancer efficacy.** As mentioned in the above experiments, PEG-$T^{ECM}$-NS with crosslinked structure has high stability during blood circulation. Upon arrival at the tumour site, the de-shielding of PEG corona from the nanoparticles triggered by tumour acidity leads to the exposure of ECM targeting peptide. Then, $T^{ECM}$-NS are anchored on the ECM and retained in the TME without penetration through interstitial space and internalization into tumour cells. The hydrophobic PPS block is oxidized to become hydrophilic and induces the disruption of nanoparticles to gradually release OLE. Free OLE enters tumour cells and induces ICD through the released HMGB1 from the dying cells. To mimic the delivery process in the TME, PEG-$T^{ECM}$-NS, or PEG-$T^{ECM}$-NS/OLE was firstly incubated at pH 6.8 with 100 μM $H_2O_2$ for about 12 h and then centrifuged to collect the supernatant. The supernatant containing OLE was used in the following experiments. ICD of tumour cells is characterized by inducing extracellular release of HMGB1 as "find me" signal and

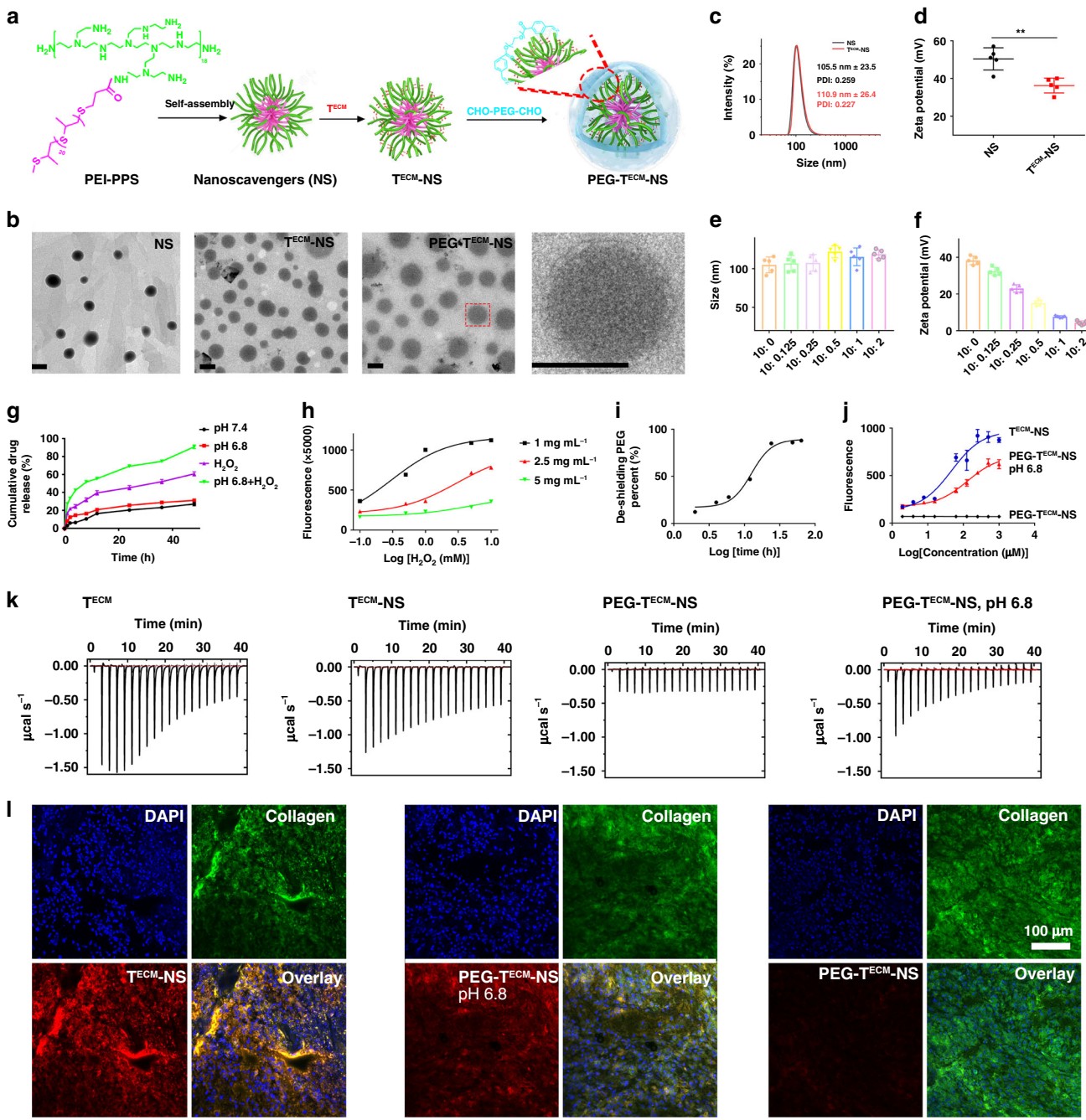

**Fig. 2 Morphology and size distributions of nanoscavengers $T^{ECM}$-NS and PEG-$T^{ECM}$-NS.** In vitro evaluation of the ability of PEG-$T^{ECM}$-NS to bind collagen and scavenge ROS. **a** Schematic illustration of the preparation of NS, $T^{ECM}$-NS and PEG-$T^{ECM}$-NS. **b** TEM images of NS, $T^{ECM}$-NS, and PEG-$T^{ECM}$-NS (Scale bar: 100 nm). Independent experiments were repeated three times. **c, d** Size distributions and zeta potentials of NS and $T^{ECM}$-NS. Error bars represent mean ± s.d. derived from $n = 5$ independent replicates. Statistical significance was evaluated using Student's unpaired $t$ test. Asterisks indicate $p$ values *$P < 0.05$, **$P < 0.01$, and ***$P < 0.001$ and n.s. represents no significant difference. **e, f** Size and zeta potential of $T^{ECM}$-NS after PEG caging at different $T^{ECM}$-NS: PEG mass ratios. Error bars represent mean ± s.d. derived from $n = 5$ independent replicates. **g** Drug release profiles of PEG-$T^{ECM}$-NS at pH 7.4 and pH 6.8 with or without 10 mM $H_2O_2$. **h** Monitoring the concentration of $H_2O_2$ in solution treated with or without PEG-$T^{ECM}$-NS. **i** PEG de-shielding rate from PEG-$T^{ECM}$-NS measured by $^1$H NMR. **j** Binding of $T^{ECM}$-NS and PEG-$T^{ECM}$-NS at pH 7.4 and pH 6.8 to rat tail collagen type I measured by solid-phase fluorescence binding assay. Independent experiments were repeated three times. **k** ITC measurements recorded by titrating the free $T^{ECM}$, $T^{ECM}$-NS, PEG-$T^{ECM}$-NS, and PEG-TECM-NS pretreated with pH 6.8 buffer against collagen in a 20-injection experiment with a time interval of 10 min between successive injections. **l** CLSM images of the slices sectioned from the 4T1 tumour pieces (1 cm$^3$) treated with $T^{ECM}$-NS and PEG-$T^{ECM}$-NS at pH 7.4 and pH 6.8 for about 4 h. Blue channel, nucleus; green channel, collagen and red channel, RB-labelled $T^{ECM}$-NS or PEG-$T^{ECM}$-NS.

cell surface expression of calreticulin (CRT) as "eat me" signal. Therefore, the ability of PEG-$T^{ECM}$-NS/OLE to induce ICD was determined by examining the HMGB1 release and CRT exposure by immunofluorescence staining. As shown in Fig. 3a, b,

pretreatment of PEG-$T^{ECM}$-NS/OLE at pH 6.8 with 100 μM $H_2O_2$ induced more extracellular release of HMGB1 than that of PEG-$T^{ECM}$-NS/OLE without pretreatment. Meanwhile, pretreatment of PEG-$T^{ECM}$-NS/OLE at pH 6.8 with 100 μM $H_2O_2$ had

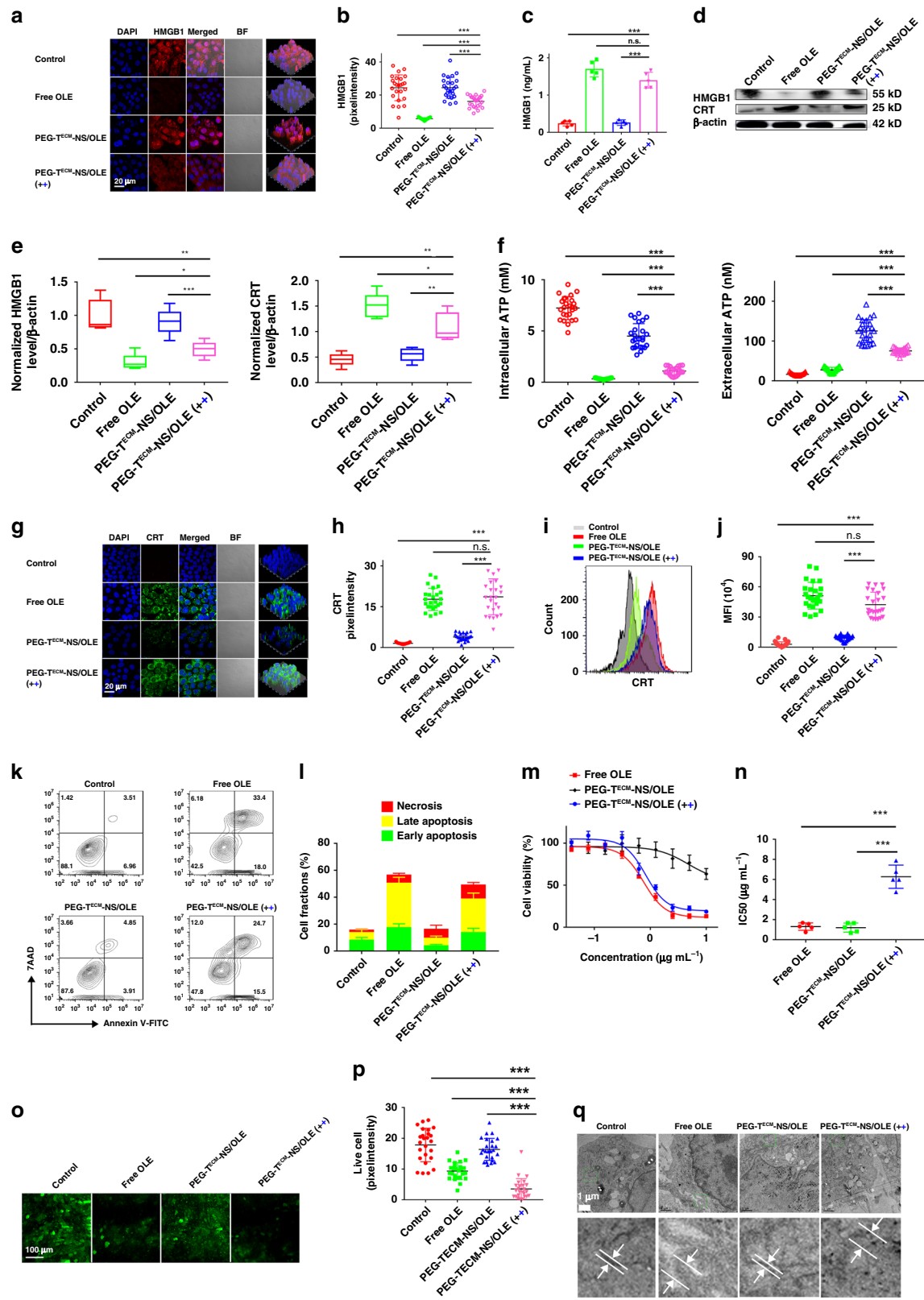

the similar ability to induce extracellular release of HMGB1 to that of free OLE. These results indicated that PEGylated T$^{ECM}$-NS/OLE with crosslinked structure could restrain drug leakage in physiological environment. The combination of the pH sensitive cleavage of the benzoic imine with de-shieldable PEG and ROS mediated hydrophilic transition of thioether induced disruption

of nanoparticles could rapidly trigger drug release. Furthermore, HMGB1 release also confirmed that free OLE and PEG-T$^{ECM}$-NS/OLE could efficiently induce immunogenicity of the tumour cells. Released HMGB1 in supernatant of 4T1 cells was measured by ELISA (Fig. 3c). Western blot was also used to evaluate the expression of HMGB1 (Fig. 3d, e). The significantly reduced

**Fig. 3 ICD and in vitro anticancer efficacy. a** HMGB1 release from 4T1 cells after incubation with free OLE and PEG-T$^{ECM}$-NS/OLE with or without pH 6.8 buffer and H$_2$O$_2$ pretreatment. Black plus represents that PEG-T$^{ECM}$-NS/OLE was pretreated with pH 6.8 buffer. Blue plus represents that PEG-T$^{ECM}$-NS/OLE was pretreated with H$_2$O$_2$. **b** Quantification of HMGB1 in 4T1 cells after different treatments ($n = 25$ independent replicates). **c** Released HMGB1 in supernatant of 4T1 cells measured by ELISA ($n = 5$ independent replicates). **d** Western blot assay of the HMGB1 expression in 4T1 cells after treatment with OLE and PEG-T$^{ECM}$-NS/OLE with or without pretreatment pH 6.8 buffer and H$_2$O$_2$. **e** Western blot grey value of HMGB1 and CRT protein intensity ($n = 5$ independent replicates). The samples were derived from the same experiment and the gels were run in parallel. For the boxplots, the middle line is the median, the lower and upper hinges correspond to the first and third quartiles, and whiskers represent ±1.5 inter-quartile range. **f** Extracellular and intracellular ATP levels in 4T1 cells after different treatments ($n = 25$ independent replicates). **g** Translocation of CRT to the surface of 4T1 cells after incubation with free OLE and PEG-T$^{ECM}$-NS/OLE with or without pretreatment (pH 6.8 buffer and H$_2$O$_2$). **h** Quantification of HMGB1 in 4T1 cells after different treatments ($n = 25$ independent replicates). **i** Flow cytometry analyses of CRT. **j** Mean fluorescence intensity (MFI) of CRT positive cells ($n = 25$ independent replicates). **k** Apoptosis induced by free OLE, PEG-T$^{ECM}$-NS/OLE, and PEG-T$^{ECM}$-NS/OLE pretreated with or without pH 6.8 buffer and H$_2$O$_2$ was investigated by Annexin V-FITC and 7-AAD double staining and analyzed via flow cytometry. **l** Quantitative analysis of corresponding cell apoptosis/necrosis percentages based on **k** ($n = 3$ independent replicates). Cytotoxicity (**m**) and IC$_{50}$ values (**n**) of free OLE, PEG-T$^{ECM}$-NS/OLE, and PEG-T$^{ECM}$-NS/OLE pretreated with or without pretreatment (pH 6.8 buffer and H$_2$O$_2$) ($n = 5$ independent replicates). **o** Fluorescence images of 4T1 cells stained by Calcein AM (green) after different treatments. **p** Quantification of Calcein AM positive cells ($n = 25$ independent replicates). **q** Representative TEM images of 4T1 cells after different treatments. All error bars represent mean ± s.d. Statistical significance was evaluated using Student's unpaired $t$ test. Asterisks indicate $p$ values *$P < 0.05$, **$P < 0.01$, and ***$P < 0.001$ and n.s. represents no significant difference.

expression of HMGB1 in intracellular 4T1 cells treated with free OLE and PEG-T$^{ECM}$-NS/OLE pretreated at pH 6.8 with 100 μM H$_2$O$_2$ indicated that considerable amount of HMGB1 was released to extracellular environment. Extracellularly secreted and intracellularly distributed ATP from tumour cells were measured (Fig. 3f). ATP secretion was studied to further verify the ability of PEG-T$^{ECM}$-NS/OLE to induce ICD. We found the intracellular ATP in the control group was significantly higher than that of PEG-T$^{ECM}$-NS/OLE pretreated at pH 6.8 with 100 μM H$_2$O$_2$. ATP secretion in the group of PEG-T$^{ECM}$-NS/OLE pretreated at pH 6.8 with 100 μM H$_2$O$_2$ was 6.93-fold higher than that of that control group (Fig. 3f). These results suggest that ICD was elicited by OLE and PEG-T$^{ECM}$-NS/OLE pretreated at pH 6.8 with 100 μM H$_2$O$_2$. CLSM and flow cytometry results showed obvious CRT translocation to the cell membrane of 4T1 cells treated with free OLE and PEG-T$^{ECM}$-NS/OLE pretreated at pH 6.8 with 100 μM H$_2$O$_2$ (Fig. 3g–j).

To further determine the therapy efficacy of PEG-T$^{ECM}$-NS/OLE as a drug delivery system, the in vitro cytotoxicity of free OLE and PEG-T$^{ECM}$-NS/OLE were evaluated and compared. As shown in Fig. 3k, l, the percentage of late apoptotic cells (Annexin V-FITC and 7-AAD double stained) was 24.7% when treated with PEG-T$^{ECM}$-NS/OLE pretreated with pH 6.8. Consistent with the results, the IC$_{50}$ values of PEG-T$^{ECM}$-NS/OLE in 4T1 cells with or without pH 6.8 buffer treatment were 0.826 and 4.75 μg mL$^{-1}$, respectively (Fig. 3m, n). In the live cell staining assay, fewer live cells were observed in the group of PEG-T$^{ECM}$-NS/OLE compared with that of PEG-T$^{ECM}$-NS/OLE (Fig. 3o, p). In mice, chemotherapy-driven ICD was related to endoplasmic reticulum stress[36–38]. The swelled endoplasmic reticulum morphology was observed in the group of free OLE and PEG-T$^{ECM}$-NS/OLE pretreated with pH 6.8 buffer, confirming ICD was elicited by OLE and PEG-T$^{ECM}$-NS/OLE (Fig. 3q).

**PEG-T$^{ECM}$-NS activated immunogenicity of tumour cells**. We further evaluated the efficacy of PEG-T$^{ECM}$-NS/OLE-induced immunogenicity of the tumour cells and the ability to turn the tumour cells into antigen-presenting cells (APC) via ICD. 4T1 tumour cells were treated with free OLE, PEG-T$^{ECM}$-NS/OLE, and PEG-T$^{ECM}$-NS/OLE pretreated at pH 6.8 with 100 μM H$_2$O$_2$ for 12 h. Then, Bone marrow dendritic cells (BMDCs) were co-cultured with pretreated 4T1 tumour cells for another 24 h. The frequency of CD80$^+$CD86$^+$ mature BMDCs after co-culture with 4T1 tumour cells pretreated with free OLE was significantly higher that of the control group, indicating that ICD from dying

4T1 cells elicited by OLE can induce DC maturation and immune response (Fig. 4a, b, and Supplementary Fig. 22). The group of PEG-T$^{ECM}$-NS/OLE pretreated at pH 6.8 with 100 μM H$_2$O$_2$ showed strong immune response with 36.9% CD80$^+$CD86$^+$ BMDCs. These results also indicated that DAMPs secreted from dying cells or exposed on the outer layer of the cell membrane can maintain their activity to facilitate immune responses in normal environment. Meanwhile, interleukin 12 (IL-12p40) and tumour necrosis factor α (TNF-α) as indicators of DC activation elicited by ICD from dying tumour cells were determined by ELISA (Fig. 4c). The result indicated that the secretion levels of IL-12p40 and TNF-α from BMDCs in groups of free OLE and pretreated PEG-T$^{ECM}$-NS/OLE (pH 6.8, 100 μM H$_2$O$_2$) were higher than those of the control group, confirming that OLE could induce strong antitumour immunity. PEG-T$^{ECM}$-NS/OLE with pre-treatment induced higher level of IL-12p40 and TNF-α than PEG-T$^{ECM}$-NS/OLE without pretreatment, suggesting that the release of OLE was significantly accelerated at acid environment in the presence of H$_2$O$_2$ due to the pH-induced PEG de-shielding and ROS induced hydrophilic transition of thioether, resulting in the disruption of nanoparticles. Mice were immunized intravenously with the tumour lyste three times. Lymph nodes were then harvested to measure the percentages of CD80$^+$CD86$^+$ DCs. The results verified that PEG-T$^{ECM}$-NS/OLE pretreated with pH 6.8 buffer and 100 μM H$_2$O$_2$ could induce the maturation of DCs (Fig. 4d). Then, we evaluated whether ROS could neutralize the stimulatory activity of DAMPs. 4T1 tumour cells were treated with free OLE, PEG-T$^{ECM}$-NS/OLE, and PEG-T$^{ECM}$-NS/OLE pretreated at pH 6.8 with 100 μM H$_2$O$_2$ for 12 h. Then, 100 μM H$_2$O$_2$ was added into the medium and 4T1 tumour cells were continually cultured for 4 h. Thereafter, BMDCs were co-cultured with pretreated 4T1 tumour cells for another 24 h. The percentage of CD80$^+$CD86$^+$ mature BMDCs was very low in the group of OLE, indicating that the HMGB1 released from dying cells was oxidized by H$_2$O$_2$ and its stimulatory activity was neutralized (Fig. 4e, f). The secretion levels of IL-12p40 and TNF-α of the group of OLE were very low (Fig. 4g). In addition, the ROS nanocanvenger T$^{ECM}$-NS/OLE involved the BMDCs/4T1 co-culture system obviously recovered the immune responses. T-cell proliferation was used to evaluate whether H$_2$O$_2$ influenced the immune responses of T cells. Splenocytes were isolated and stimulated with concanavalin A (ConA) and then incubated with or without 100 μM H$_2$O$_2$ or PEG-T$^{ECM}$-NS. T-cell proliferation was measured by carboxyfluorescein (CFSE) dilution (Fig. 4h, i). The proliferation of CD8$^+$ T cells was significantly inhibited after H$_2$O$_2$ treatment. In the group of PEG-T$^{ECM}$-NS (pH 6.8, 100 μM

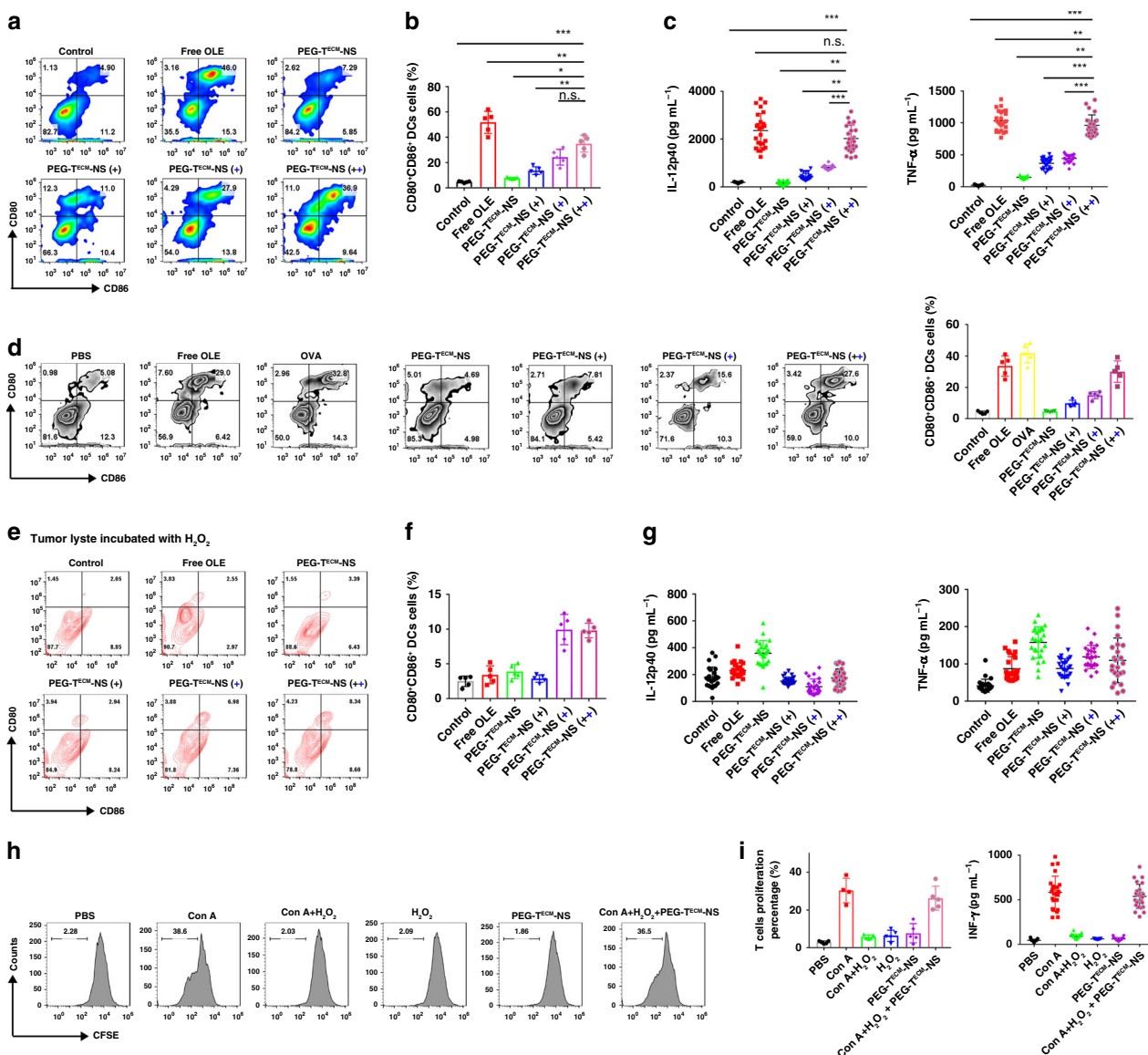

**Fig. 4 ICD-associated DAMPs from 4T1 cells induce the maturation of BMDCs in vitro. a** 4T1 cells were incubated with free OLE and PEG-T$^{ECM}$-NS/OLE with or without pretreatment pH 6.8 buffer and H$_2$O$_2$, followed by co-culture with BMDCs. The cells were then labelled with CD86 and CD80, and mature DCs were measured using flow cytometry. Black plus represents that PEG-T$^{ECM}$-NS/OLE was pretreated with pH 6.8 buffer. Blue plus represents that PEG-T$^{ECM}$-NS/OLE was pretreated with H$_2$O$_2$. **b** Quantitative analysis of CD80+CD86+ DCs ($n = 5$ independent replicates). **c** Quantification of secretion of IL-12p40 and TNF-α in DC suspensions ($n = 25$ independent replicates). **d** 4T1 cells were incubated with free OLE and PEG-T$^{ECM}$-NS/OLE with or without pretreatment (pH 6.8 buffer (4 h) and H$_2$O$_2$ (24 h)) to obtain the tumour lyste. Then mice were immunized intravenously with the tumour lyste three times. Seven days after the third immunization, lymph nodes were harvested to measure the percentage of CD80+CD86+ DCs by flow cytometry (left). Quantification of CD80+CD86+ DCs (right) ($n = 5$ independent replicates), OVA served as the control antigen. **e, f** The supernatants of 4T1 cells incubated with free OLE and PEG-T$^{ECM}$-NS/OLE with or without pH 6.8 buffer and H$_2$O$_2$ pretreatment were oxidized by H$_2$O$_2$. Then BMDCs were co-cultured with oxidized supernatants from 4T1 cells for 3 days with different treatments. Representative flow cytometric plots of mature BMDCs (**e**) and quantification result (**f**) ($n = 5$ independent replicates). **g** Quantification of secretion of IL-12p40 and TNF-α in DC suspensions ($n = 25$ independent replicates). **h** T cells were acquired from spleens and labelled with CFSE. T cells were cultured with PEG-T$^{ECM}$-NS, 100 µM H$_2$O$_2$, and PEG-T$^{ECM}$-NS with 100 µM H$_2$O$_2$. The proliferated T cells were measured using flow cytometry. **i** Quantification of T-cell proliferation ($n = 5$ independent replicates) and IFN-γ in T-cell culture medium ($n = 25$ independent replicates). All error bars represent mean ± s.d. Statistical significance was evaluated using Student's unpaired t test. Asterisks indicate p values *P < 0.05, **P < 0.01, and ***P < 0.001 and n.s. represents no significant difference.

H$_2$O$_2$), the suppressive effect of H$_2$O$_2$ on T-cell proliferation of was partially abrogated, showing that PEG-T$^{ECM}$-NS can scavenge extracellular ROS to extend the survival of T cells and recover immune responses. We also found H$_2$O$_2$ concentration dependent inhibition of T-cell proliferation (Supplementary Figs. 23 and 24).

**In vivo antitumour activity of PEG-T$^{ECM}$-NS/OLE.** The in vivo distribution of Cy5.5-labelled PEG-T$^{ECM}$-NS in 4T1 tumour-bearing mice was evaluated. As shown in Fig. 5a, b, PEG-T$^{ECM}$-NS with crosslinked structure exhibited higher tumour fluorescence signal, compared with the nanoparticles without crosslinker. We also quantified the concentrations of OLE in the blood

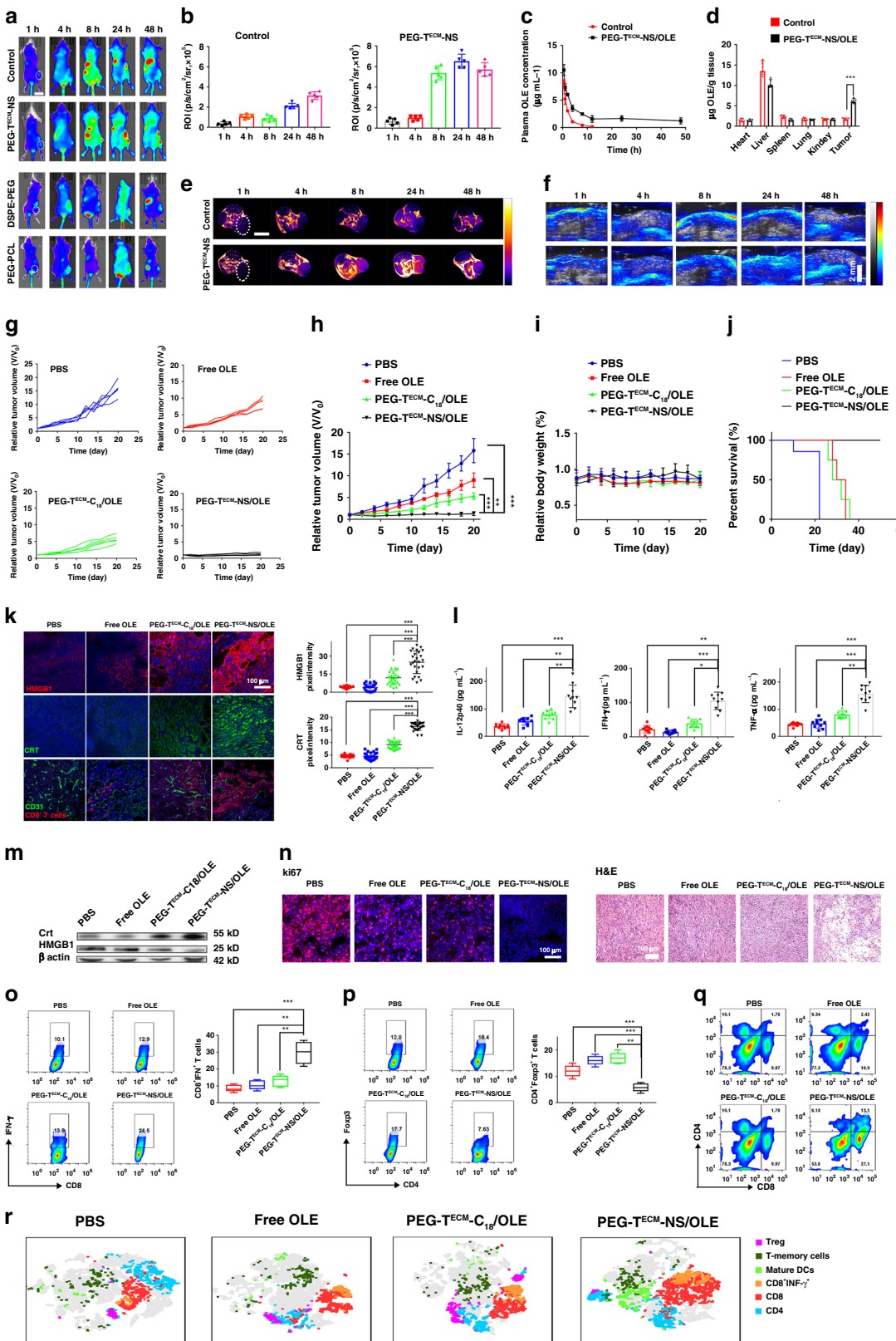

after intravenous administration of the samples (Fig. 5c). The total OLE amount in tumour was presented in Fig. 5d. The in vivo distribution of Gd-labelled PEG-T$^{ECM}$-NS was investigated by magnetic resonance imaging (MRI) and ICG-labelled PEG-T$^{ECM}$-NS was detected by photoacoustic imaging (Fig. 5e, Supplementary Figs. 25 and 5f). The results showed crosslinked PEG-T$^{ECM}$-NS

with prolonged blood circulation time and enhanced tumour accumulation.

The ability of PEG-T$^{ECM}$-NS/OLE to inhibit tumour growth was evaluated in 4T1 tumour-bearing BALB/C mice. PEG-T$^{ECM}$-C$_{18}$/OLE without ROS scavenging ability as the control group was also evaluated. As shown in Fig. 5g, h, free OLE showed a

**Fig. 5 In vivo antitumour activity of PEG-T$^{ECM}$-NS/OLE in Balb/c mice bearing 4T1 breast tumour. a** In vivo fluorescence imaging after intravenous injection of Cy5.5-labelled PEG-T$^{ECM}$-NS without crosslinked structure (as control group), PEG-T$^{ECM}$-NS, DSPE-PEG, and PEG-PCL. **b** Quantitative analysis of fluorescence signal ($n = 5$ biologically independent mice), scale bar 2 cm. **c** Blood clearance curves of OLE after intravenous injection of different formulas ($n = 5$ biologically independent mice). **d** Biodistribution of OLE in tumour after intravenous administration of different treatments. Estimation of OLE in tissues was performed using high performance liquid chromatography (HPLC) ($n = 5$ biologically independent mice). **e** $T_1$-weighted MRI of the 4T1 tumour-bearing mice after tail vein injection of PEG-T$^{ECM}$-NS without crosslinked structure as control group and PEG-T$^{ECM}$-NS modified with Gd at a dose of 10 mg kg$^{-1}$ body weight, scale bar 1 cm. **f** PA imaging of the 4T1 tumour mice after different treatments, scale bar 1 cm. The individual (**g**) and averaged (**h**) 4T1 tumour growth curves after different treatments ($n = 5$ biologically independent mice). **i** Body weight of 4T1 tumour mice treated with different formulations ($n = 5$ biologically independent mice). **j** Cumulative survival curves of tumour mice. **k** Immunofluorescence of HMGB1, CRT, and CD8$^+$ T cells in the tumour tissues after different treatments (left) and quantitative analyses (right) ($n = 25$ independent replicates). **l** Quantification of secretion of IL-12p40, IFN-γ, and TNF-α in sera from mice isolated from groups of mice on day 7 ($n = 10$ biologically independent mice). **m** Western blot analysis of HMGB1 and CRT expression levels in the tumour tissues. The samples were derived from the same experiment and the gels were run in parallel. **n** Ki67 staining and H&E staining of the tumour tissues with different treatments. **o** The determination of IFN-γ positive CD8$^+$ T cells (CD8$^+$IFN-γ$^+$ T cells) within tumour by flow cytometry (left) and quantitative analysis (right) ($n = 5$ biologically independent mice). **p** The percentage of Tregs (CD4$^+$Foxp3$^+$ T cells) within tumour by flow cytometry (left) and quantitative analysis (right) ($n = 5$ biologically independent mice). For the boxplots (**o**, **p**), the middle line is the median, the lower and upper hinges correspond to the first and third quartiles, and whiskers represent ±1.5 inter-quartile range. **q** CD8$^+$ T cells in 4T1 tumour analyzed by flow cytometry. **r** T-distributed stochastic neighbour embedding (t-SNE) visualization of clustering of representative markers of cells from tumour detected by flow cytometry, each dot corresponds to one single cell. All error bars represent mean ± s.d. Statistical significance was evaluated using Student's unpaired $t$ test. Asterisks indicate $p$ values *$P < 0.05$, **$P < 0.01$, and ***$P < 0.001$ and n.s. represents no significant difference.

moderate reduction of tumour growth by 25% as compared to the PBS group. PEG-T$^{ECM}$-C$_{18}$/OLE had better antitumour effect than free OLE as evidenced by 55% reduction in tumour growth compared with the PBS group. Notably, PEG-T$^{ECM}$-NS/OLE achieved 92% reduction in the tumour volume. As shown in Fig. 5i, no significant change in the body weight of mice treated with PEG-T$^{ECM}$-NS/OLE was observed during treatment, verifying the biocompatibility and safety of PEG-T$^{ECM}$-NS/OLE. The survival time mice in each group correlated well with the tumour inhibition result (Fig. 5j). Dying tumour cells generated HMGB1 through PEG-T$^{ECM}$-NS/OLE-induced ICD. The immunostimulatory activity of HMGB1 was activated because the ROS in the TME was effectively cleared by PEG-T$^{ECM}$-NS/OLE nanoscavenger. The immunostimulatory activity of HMGB1 elicited by PEG-T$^{ECM}$-NS/OLE was evaluated by immunofluorescence assay (Fig. 5k). PEG-T$^{ECM}$-NS/OLE treatment with ROS scavenging ability can evoke large amount of HMGB1 in the extracellular environment compared with PEG-T$^{ECM}$-C$_{18}$/OLE without ROS scavenging ability. CRT exposure induced by PEG-T$^{ECM}$-NS/OLE treatment was much higher than that of the other groups, suggesting that PEG-T$^{ECM}$-NS/OLE realized effective ICD. Modulating the level of extracellular ROS in the TME has been reported to extend survival of T cells[13,15,29,39]. PEG-T$^{ECM}$-NS/OLE treatment increased CD8$^+$ T-cell permeation in 4T1 mouse model as compared to PEG-T$^{ECM}$-C$_{18}$/OLE. As shown in Fig. 5k, the immunofluorescence staining results indicated that PEG-T$^{ECM}$-NS/OLE significantly increased the infiltration of CD8$^+$ T cells in the tumour region. To evaluate the antitumour immunity, peripheral blood serum was harvested from 4T1 tumour mice with different treatments on day 7 (after the first treatment) and analysed by ELISA. As shown in Fig. 5l, the production of IL-12p40, TNF-α, and interferon-gamma (IFN-γ) were increased in the mice treated with PEG-T$^{ECM}$-NS/OLE. The result indicated that PEG-T$^{ECM}$-NS/OLE can elicit effective ICD. As shown in Fig. 5m, the expression of HMGB1 and CRT protein in cells treated with the PEG-T$^{ECM}$-NS/OLE was obviously elevated, which agreed well with the result in Fig. 5k. Compared with other groups, administration of PEG-T$^{ECM}$-NS/OLE markedly reduced the number of Ki67 proliferating tumour cells and increased the number of dUTP nick-end labelling (TUNEL)-positive tumour cells (Fig. 5n). Moreover, hematoxylin and eosin (H&E) staining of the tumour tissue of mice treated with PEG-T$^{ECM}$-NS/OLE showed extensive tumour cell death. No observable damage to normal tissues and major organs was found,

suggesting the biocompatibility of PEG-T$^{ECM}$-NS/OLE (Supplementary Fig. 26). As shown in Fig. 5o–q and Supplementary Fig. 27, compared with the control groups, the increased number of IFN-γ expressing CD8$^+$ T cells and decreased number of Foxp3$^+$CD4$^+$ T cells in the group of PEG-T$^{ECM}$-NS/OLE further elicited powerful antitumour T-cell response. The t-SNE analysis of tumour tissues from mice after different treatments indicated increased IFN-γ$^+$CD8$^+$T cells, CD8$^+$ T cells, and T memory cells infiltration into tumours treated PEG-T$^{ECM}$-NS/OLE (Fig. 5r).

Then, we established an orthotopic colorectal cancer model with CT26 tumour cells. Intraperitoneal injection of L-012 into orthotopic colorectal tumour mice allowed to detect ROS scavenging by nanoparticles at the tumour site. PEG-T$^{ECM}$-NS treatment had reduced bioluminescence signal compared to PEG-T$^{ECM}$-NS without ROS scavenging ability (Fig. 6a). Quantification of bioluminescence signals indicated the ROS levels in the group of PEG-T$^{ECM}$-NS without ROS scavenging ability was 3.41- and 2.52-fold higher than those of PEG-T$^{ECM}$-NS at 24 and 72 h, respectively (Fig. 6b). We investigated the tumour inhibition of PEG-T$^{ECM}$-NS/OLE in orthotopic CT26 tumour mice. As shown in Fig. 6c, PEG-T$^{ECM}$-NS/OLE showed 91.9% reduction in the colon tumour. PEG-T$^{ECM}$-C$_{18}$/OLE without ROS scavenging ability achieved 68.8% reduction. As shown in Fig. 6d, no significant change in the body weight of mice treated with PEG-T$^{ECM}$-NS/OLE was observed during treatment, verifying the biocompatibility and safety of PEG-T$^{ECM}$-NS/OLE. We performed immunofluorescence analysis to study the expression of HMGB1, CRT, and CD8$^+$ T cells in the tumour tissues after different treatments. We observed detectable increase of HMGB1 and CRT in the tumour from the group of PEG-T$^{ECM}$-NS/OLE, demonstrating that ICD was efficiently elicited (Fig. 6e). PEG-T$^{ECM}$-NS/OLE treatment increased CD8$^+$ T-cell permeation as compared to PEG-T$^{ECM}$-C18/OLE. The expression levels of HMGB1 and CRT protein in orthotopic CT26 tumour mice treated with the PEG-TECM-NS/OLE were obviously elevated, which agreed well with the immunofluorescence analysis results (Fig. 6f). Caspase-3 and H&E staining of the tumour tissue of mice treated with PEG-T$^{ECM}$-NS/OLE showed extensive tumour cell death (Fig. 6g). The levels of cytokine secretion in serum were measured by ELISA to monitor the nanoscavenger induced immune response. PEG-T$^{ECM}$-NS/OLE group displayed a significant increase in IL-12p40, IFN-γ, and TNF-α expressions in the serum, which confirmed that nanoscanvenger effectively induced ICD and improved the immune responses (Fig. 6h). The proportion of CD8$^+$ T cells in

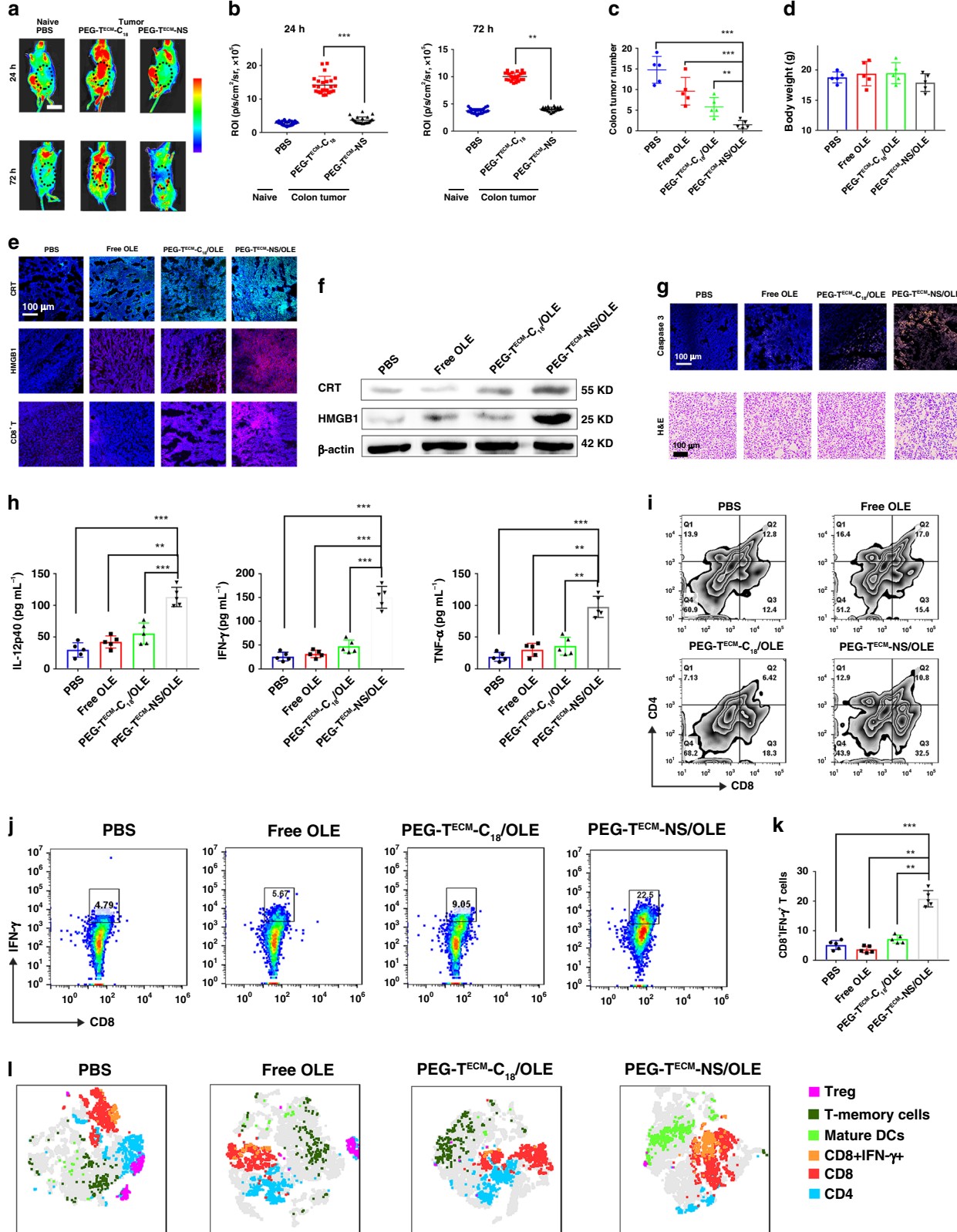

the tumours of mice treated with PEG-T$^{ECM}$-NS/OLE showed an obviously increase over PEG-T$^{ECM}$-C$_{18}$/OLE without ROS scavenging ability (Fig. 6i). Furthermore, PEG-T$^{ECM}$-NS/OLE induced a higher proportion of IFN-γ$^+$CD8$^+$ T cells in spleens than that of PEG-T$^{ECM}$-C$_{18}$/OLE (Fig. 6j, k). t-SNE analysis of tumour tissues from mice after different treatments indicated increased

IFN-γ$^+$CD8$^+$ T cells, CD8$^+$ T cells and T memory cells infiltration into tumours treated PEG-T$^{ECM}$-NS/OLE (Fig. 6l).

Extracellular ROS abolishes immune response elicited by ICD and functionally suppresses T cells activation. Therefore, we employed a nanoscavenger to sweep away the extracellular ROS at the tumour site to enhance immunotherapy by relieving the immunosuppressive

**Fig. 6 In vivo antitumour and ROS scavenging activity of PEG-T^ECM-NS/OLE in CT26 orthotopic colorectal cancer model. a** In vivo chemiluminescence imaging of ROS in tumour microenvironment by L-012. **b** Quantification of L-012 chemiluminescence ($n = 5$ biologically independent mice). **c** Colon tumour numbers after different treatment ($n = 5$ biologically independent mice). **d** Body weight of orthotopic colorectal cancer mice after different treatments ($n = 5$ biologically independent mice). **e** Immunofluorescence of HMGB1, CRT, and CD8$^+$ T cells in the tumour tissues after different treatments. **f** Western blot analysis of HMGB1 and CRT expression levels in the tumour tissues. The samples were derived from the same experiment and the gels were run in parallel. **g** Caspase-3 staining and H&E staining of the tumour tissues with different treatments. **h** Quantification of secretion of IL-12p40, IFN-γ, and TNF-α in sera from mice ($n = 5$ biologically independent mice). **i** CD8$^+$ T cells in 4T1 tumour analyzed by flow cytometry. **j** The determination of IFN-γ positive CD8$^+$ T cells (CD8$^+$IFN-γ$^+$ T cells) within tumour by flow cytometry. **k** Quantitative analysis of IFN-γ$^+$CD8$^+$ T cells ($n = 5$ biologically independent mice). **l** T-distributed stochastic neighbour embedding (t-SNE) visualization of clustering of representative markers of cells from tumour detected by flow cytometry, each dot corresponds to one single cell. All error bars represent mean ± s.d. Statistical significance was evaluated using Student's unpaired $t$ test. Places needing multiple comparisons were evaluated by one-way ANOVA. Asterisks indicate $p$ values *$P < 0.05$, **$P < 0.01$, and ***$P < 0.001$ and n.s. represents no significant difference.

ICD and increasing infiltration of CD8$^+$ T cells. Overall, these results suggest that scavenging extracellular ROS is a promising strategy to increase the efficacy of cancer immunotherapy.

## Methods

**Preparation and characterization of nanoscavengers.** PEI-PPS was prepared by a solvent displacement method. Typically, the solution of dimethyl sulfoxide (DMSO) containing 10 mg PEI-PPS polymer was added dropwise to water. The mixture was dialyzed to remove DMSO at room temperature. Then T^ECM, EDC, and NHS were added to PEI-PPS nanoparticle solution to obtain T^ECM modified nanoparticles (T^ECM-NS). After stirring for 24 h, the mixture was dialyzed. Then the aqueous solution of CHO-PEG-CHO was added to T^ECM-NS solution to coat PEG on the surface of T^ECM-NS through benzoic bonds.

**ELISA protocol to determine collagen binding constant.** Collagen-coated plates were prepared by adding 50 μL of a 1:100 dilution of rat tail collagen type I (2.78 mg mL$^{-1}$ in 20 mM acetic acid) in TBS (50 mM Tris-HCl, 150 mM NaCl, pH = 7.4) to rows A–D on a 96-well plate (4 plates, one per construct). The plates were incubated overnight at 4 °C on a rotating table, then inverted, emptied, and washed with TBS (3 × 300 μL). All the wells (i.e. rows A–H) were then incubated with 100 μL of TBS containing 5% BSA for 2 h at 37 °C, and subsequently washed with 3 × 300 μL of TBS containing 0.1% Tween-20. The plates were incubated with the PEG-T^ECM-NS for 3 h. The fluorescence was measured.

**Cell apoptosis detection.** 4T1 Cells were seeded in six well plates. The supernatant of PEG-T^ECM-NS pretreated with pH 6.8 buffer and 10 mM H$_2$O$_2$ after centrifugation were added and incubated with cells. Then 4T1 cells were stained with Annexin V-FITC (5 μL for one sample)/7-AAD (5 μL for one sample) (FITC Annexin V Apoptosis Detection Kit with 7-AAD, Biolegend, Catalog number 640922) for 30 min at room temperature and then added 500 μL binding buffer to analyze by a flow cytometry.

**Animal model.** All animal experiments were performed under a National Institutes of Health Animal Care and Use Committee (NIHACUC) approved protocol. BALB/C mice (Harlan, Indianapolis, IN) were subcutaneously implanted with 1 × 106 4T1 cells. 4T1 cells (ATCC) were authenticated for mycoplasma.

**BMDC culture and stimulation.** BMDCs were isolated from the femur of BALB/C mice. The bone marrow cells were harvested and treated with ACK lysis buffer for 5 min. The cells were washed twice with PBS. The bone marrow was cultured in medium with GM-CSF (10 ng mL$^{-1}$) and IL-4 (10 ng mL$^{-1}$) at 37 °C for 7 days to acquire immature DC. 4T1 cells were pretreated with free OLE, PEG-T^ECM-NS/OLE, and PEG-T^ECM-NS/OLE pretreated with pH 6.8 buffer and 100 μM H$_2$O$_2$. Then, immature BMDCs (1 × 10$^6$) were co-cultured with 4T1 cells (1 × 10$^5$) for 24 h. After various treatments, DCs stained with antibodies (anti-CD11c-APC, anti-CD80-FITC, and anti-CD86-PE) for 30 min at 4 °C and measured by a flow cytometer. Anti-CD11c-APC-Cy7, anti-CD80-FITC, and anti-CD86-PE were diluted according to the manufacturer's direction (1:500).

**T-cell proliferation assay.** Splenocytes were isolated and incubated for 4 h. We collected the cells and labelled with CFSE (1 mM) for 5 min. Then, 0.6 mg mL$^{-1}$ concanavalin A (ConA) and H$_2$O$_2$ were added. After 3 days, the CFSE signal of gated splenocytes was analyzed by a flow cytometer.

**Tumour-infiltrating lymphocytes analysis.** The harvested tumours were explanted and cut into small pieces. Then, the small pieces were immersed in 5 mL collagenase IV (1 mg mL$^{-1}$) with 0.2 mg mL$^{-1}$ DNase I for 1 h at 37 °C. Suspensions were filtered and the single cells were stained with fluorescently labelled antibodies (CD4$^+$CD25$^+$Foxp3$^+$ Tregs, CD4$^-$CD8$^+$ T cells, CD8$^+$IFN-γ$^+$ T cells, CD8$^+$CD44$^+$CD62L$^+$ memory T cells). Anti-CD4-PE-Cy5, anti-CD8-PE-Cy7, and anti-CD25-Alexa Fluor ®700, anti-CD62L-APC, anti-CD44-PerCP, anti-Foxp3-PE, and anti-IFN-γ-PerCP/Cy5.5 were diluted according to the manufacturer's direction (1:500).

**In vivo PET imaging.** Firstly, we prepared DFO modified PEG-T^ECM-NS with crosslinked structure (DFO-PEG-T^ECM-NS). As a control, we also prepared N-PEG-T^ECM-NS without crosslinked structure DFO-N-PEG-T^ECM-NS. Zr$^{89}$-DFO-PEG-T^ECM-NS, and Zr$^{89}$-DFO-N-PEG-T^ECM-NS were intravenously injected into the 4T1 tumour-bearing mice. An Inveon small-animal PET scanner (Siemens, Erlangen, Germany) was used to acquire whole-body PET images at predetermined time points after injection.

**In vivo MR imaging.** Firstly, we prepared DOTA modified PEG-T^ECM-NS with crosslinked structure (DOTA-PEG-T^ECM-NS). As a control, we also prepared N-PEG-T^ECM-NS without crosslinked structure DOTA-N-PEG-T^ECM-NS. Gd$^{3+}$-DOTA-PEG-T^ECM-NS and Gd$^{3+}$-DOT -N-PEG-T^ECM-NS were intravenously injected into the 4T1 tumour-bearing mice. A high magnetic field micro-MR scanner (7.0 T, Bruker, Pharmascan) was used for the scanning at indicated time points after injection.

**In vivo antitumour therapy.** BALB/C mice were inoculated with 4T1 tumour cells. The mice were randomly grouped and treated with PBS, free OLE, PEG-T^ECM-C$_{18}$/OEL, and PEG-T^ECM-NS/OLE (OLE 1 mg kg$^{-1}$ for a total dose). Then the tumour volume was calculated using the formula: tumour volume (mm$^3$) = (length × width$^2$) × 1/2.

**Orthotopic CT26 tumour model establishment.** Mice were anesthetized with isoflurane. A small incision was made to exteriorize the caecum. CT26 cell suspension (1.0 × 10$^6$) was injected into the caecum wall. Then the incision was sutured. The mice were randomly grouped and treated with PBS, free OLE, PEG-TECM-C18/OEL, and PEG-TECM-NS/OLE (total OLE dose: 1 mg kg$^{-1}$). Mice were sacrificed by cervical dislocation after treatment. Spleen, tumour, and main organs were collected for immunofluorescence and flow cytometry analysis.

**Statistics and reproducibility.** All representative images were performed a minimum of three replicates in independent experiments with similar results.

**Statistical analysis.** All values are expressed as mean ± standard deviation (SD) from. The significant differences between two groups were evaluated by Student's unpaired $t$ test. Statistical significance is displayed as: N.S., not significant; *$P < 0.05$; **$P < 0.01$; ***$P < 0.001$.

**Reporting summary.** Further information on research design is available in the Nature Research Reporting Summary linked to this article.

## Data availability

The experimental data that support the findings in the current study are available for research purposes within the paper and its Supplementary Information from the corresponding authors on reasonable request. Source data are provided with this paper.

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

## Acknowledgements

This research was supported by National Natural Science Foundation of China (NSFC) projects (Grant number: 201874024), and the intramural research program of the National Institute of Biomedical Imaging and Bioengineering (NIBIB), National Institutes of Health (NIH).

## Author contributions

H.D. and X.C. conceived and designed the research. H.D. and Z.Z. performed the MRI studies. H.D., W.Y., Y.M., R.T., and L.L. performed the in vitro experiments and analysed the data. H.D. and W.Y. performed the in vivo experiments and analysed the data. H.D., J.S., and X.C. co-wrote the paper.

## Competing interests

The authors declare no competing interests.
