## [Peer Review File · Nature Communications]

REVIEWER COMMENTS

Reviewer #1 (Remarks to the Author): Expert in nanotechnology and immunotherapy

In this manuscript, the authors develop a tumour extracellular matrix (ECM) targeting ROS nanoscavenger to sweep away the ROS from tumour microenvironment to relieve the immunosuppressive ICD for cancer immunotherapy. Some issues in this paper should be addressed before publication in Nature Communications

HMGB1 is an autophagy sensor in the presence of oxidative stress. In figure 3, the authors should investigate the effect of H₂O₂ to cancer cells. It has been reported that hydrogen peroxide stimulates different cells to actively release HMGB1 (Journal of leukocyte biology, 81(3), 741; Antioxidants & redox signaling 15.8 (2011): 2185-2195.). The HMGB1 release from cancer cells treated with PEG-TECM-NS/OLE at pH 6.8 with 100 μ M H₂O₂ in figure 3 could be result from the H₂O₂.

Figure 4, the authors should study the BMDCs activation after incubated with different formulations without the 4T1 cancer cells, to confirm that the activation of DCs is induced by the ICD of cancer cells or the formulations. As I have mentioned before, H₂O₂ at certain concentration could also activate the DCs (Free Radic Biol Med. 2012;52(3):635;).

Fig. 3f, Extracellularly secreted and intracellularly distributed ATP from tumour cells should be explained.

On page 8, Fig 4h-k should be fig 3h-k. In addition, fig 3k dose not discussed in the main text. Fig. 3g does not seem to be mentioned in the main text.

In page 10, the authors described "4T1 tumour cells were treated with free OLE, PEG-TECM-NS/OLE and PEGTECM-NS/OLE pretreated at pH 6.8 with 10 mM H₂O₂ for 12 h. Why this experiment use 10mM H₂O₂ needs explanation. Because the authors used 100 μ M H₂O₂ in all other experiments.

In figure 8a, the authors should confirm the contrast is consistent in all imaging. The scale intensity bar should be added.

In figure 8, it would be interesting to test if the ROS within the tumor was reduced after treatment.

Please clearly state sample size and how many times are repeated for each experiment. Please add the statistical information in many figures. Please add the scale bar in figures (e.g. fig 5k)

Reviewer #2 (Remarks to the Author): Expert in ROS scavenging and nanotherapy

In the manuscript, the authors prepared a tumor extracellular matrix (ECM) targeting ROS nanoscavenger. Dual-benzaldehyde terminated polyethylene glycol as a caging polymer was

introduced to construct the cross-linked “stealth” delivery system (PEG-TECM-NS) with pH sensitive imine bonds. This intelligent nanoscavenger that can sweep away the ROS from tumor microenvironment to relieve the immunosuppressive ICD elicited by specific chemotherapy based on oleandrin (OLE) anticancer drug and prolong the survival of T cells for “personalized” cancer immunotherapy. When arriving at the tumor site, the de-shielding of PEG corona triggered by tumor acidity leads to the exposure of ECM targeting peptide and anchor on the ECM, allowing TECM-NS to continuously scavenge extracellular ROS. Meantime, TECM-NS was oxidized by ROS inducing the disruption to release OLE. Free OLE was internalized into tumor cells and induced ICD through the release of HMGB1 from the dying cells. Therefore, the nanoscavenger anchored on the ECM to sweep away the ROS from tumor microenvironment to relieve the immunosuppressive ICD elicited by specific chemotherapy and prolong the survival of T cells for personalized cancer immunotherapy. The nanoscavenger presented by authors is interesting, and the manuscript is well written. This work appears suitable for publication in Nature Communications after minor revision.

1. In the manuscript, the authors said that “Recent studies revealed that the immunogenic cell death (ICD) elicited by specific chemotherapy or radiotherapy makes their corpses ‘visible’ to dendritic cells (DCs) that present antigens to T cells with specific antitumor immune responses, which then control residual tumor cells” Could you give the detail examples of the ICD elicited by chemotherapy.

2. The in vitro accumulated drug release was very important. The results of drug release can confirm the stability of cross-linked nanoparticles and the sensitive drug release. The release rate of OLE could be significantly enhanced in pH 6.8 with 10 mM H₂O₂. To show the data more clearly, the authors should give the detail data of cumulative drug release in Figure 2g.

3. The binding affinity of PEG-TECM-NS and TECM-NS on collagen was measured by enzyme-linked immunosorbent assay (ELISA) using biotinylated nanoparticle and substrates coated with rat tail collagen type I. Please provide the data of 4T1 tumors pieces treated with free collagen targeting peptides

4. Free OLE was internalized into tumor cells and induced ICD. ICD of tumor cells is characterized by inducing extracellular release of HMGB1 as “find me signals” and cell surface expression of calreticulin (CRT) as “eat me signals”. The release of ATP was also a characteristic of ICD. Extracellular secreted and intracellular distributed ATP from tumor cells was measured as shown in Fig. 3f. Please explain the difference between the groups

5. To evaluate the efficacy of PEG-TECM-NS/OLE induced immunogenicity of the tumor cells and turned the tumor cells into antigen-presenting cell (APC) via ICD, the cell surface specific expression of CD80 and CD86 marker of bone marrow dendritic cells (BMDCs) separated from BALB/c mice maturation induced by ICD was investigated using flow cytometry. Please provide the detail experiment process.

6. Splenocytes were isolated and stimulated with concanavalin A (ConA) and then incubated with or without 100 μM H₂O₂ or PEG-TECM-NS. T-cell proliferation was measured by carboxyfluorescein (CFSE) dilution. What’s the function of ConA. Please give some literatures.

7. In the manuscript, the authors also examined the inhibition of T cell proliferation under different concentration of H₂O₂ in vitro. Please provide the quantitative results.

8. Could you give the quantitative analysis of the MRI results in Fig. 5e.

Response to reviewers' comments

Reviewer #1 (Remarks to the Author): Expert in nanotechnology and immunotherapy

In this manuscript, the authors develop a tumour extracellular matrix (ECM) targeting ROS nanoscavenger to sweep away the ROS from tumour microenvironment to relieve the immunosuppressive ICD for cancer immunotherapy. Some issues in this paper should be addressed before publication in Nature Communications.

Reply- Thanks a lot for your constructive comments and suggestions. We have made proper changes and supplemented additional experiments according to the comments.

Question 1. HMGB1 is an autophagy sensor in the presence of oxidative stress. In figure 3, the authors should investigate the effect of H₂O₂ to cancer cells. It has been reported that hydrogen peroxide stimulates different cells to actively release HMGB1 (Journal of leukocyte biology, 81(3), 741; Antioxidants & redox signaling 15.8 (2011): 2185-2195.). The HMGB1 release from cancer cells treated with PEG-T^{ECM}-NS/OLE at pH 6.8 with 100 μM H₂O₂ in figure 3 could be result from the H₂O₂.

Reply: In Figure 3, to mimic the delivery process in the tumour microenvironment (TME), PEG-T^{ECM}-NS or PEG-T^{ECM}-NS/OLE was firstly incubated at pH 6.8 with 100 μM H₂O₂ for about 12 h and then centrifuged to collect the supernatant. The supernatant containing OLE was used in the experiments. The residual H₂O₂ concentration in the supernatant was detected using a H₂O₂ assay kit (Figure R1). There was a sharp decrease in fluorescence at 562 nm in the supernatant, suggesting the consumption of H₂O₂ by nanoparticles. Meanwhile, the result also indicated the strong H₂O₂ scavenging activity of PEG-T^{ECM}-NS. Therefore, the HMGB1 released from cancer cells treated with PEG-T^{ECM}-NS/OLE at pH 6.8 with 100 μM H₂O₂ in figure 3 was not a result of the residual H₂O₂.

Figure R1 A H₂O₂ activity kit was used to monitor the concentration of residual H₂O₂ in the supernatant.

Question 2. Figure 4, the authors should study the BMDCs activation after incubated with different formulations without the 4T1 cancer cells, to confirm that the activation of DCs is

induced by the ICD of cancer cells or the formulations. As I have mentioned before, H₂O₂ at certain concentration could also activate the DCs (Free Radic Biol Med. 2012;52(3):635;).

Reply: We evaluated the BMDCs activation after incubation with different formulations without the 4T1 cancer cells (Figure R2). The frequency of CD80⁺CD86⁺ mature BMDCs after culture with free OLE and PEG-T^{ECM}-NS/OLE with or without pretreatment pH 6.8 buffer and H₂O₂ were similar to the control group, indicating that nanoparticles could not directly induce DCs maturation and immune response. As mentioned above, the residual H₂O₂ in the supernatant has been drained by nanoparticles. The activation of BMDCs was thus not from the residual H₂O₂.

Figure R2 BMDCs were incubated with free OLE and PEG-T^{ECM}-NS/OLE with or without pretreatment pH 6.8 buffer and H₂O₂. The cells were then labeled with CD86 and CD80, and mature DCs were measured using flow cytometry (left). Black (+) represents that PEG-T^{ECM}-NS/OLE was pretreated with pH 6.8 buffer. Blue (+) represents that PEG-T^{ECM}-NS/OLE was pretreated with H₂O₂. (b) Quantitative analysis of CD80⁺CD86⁺ DCs (right).

Question 3. Fig. 3f, Extracellularly secreted and intracellularly distributed ATP from tumour cells should be explained.

Reply: We have added detailed explanation of the extracellularly secreted and intracellularly distributed ATP from tumour cells. ATP secretion was evaluated by ATP assay to further verify the ICD induction property of PEG-T^{ECM}-NS/OLE. We found the intracellular ATP in the control group was significantly higher than that of PEG-T^{ECM}-NS/OLE pretreated at pH 6.8 with 100 μM H₂O₂ group. The ATP secretion in the cell culture medium of PEG-T^{ECM}-NS/OLE pretreated at pH 6.8 with 100 μM H₂O₂ group was 6.93-fold higher than that of control group (Figure 3f). These results suggest that ICD was elicited by OLE and PEG-T^{ECM}-NS/OLE pretreated at pH 6.8 with 100 μM H₂O₂.

Question 4. On page 8, Fig 4h-k should be fig 3h-k. In addition, fig 3k dose not discussed in the main text. Fig. 3g does not seem to be mentioned in the main text.

Reply: We have rephrased the figure legends and highlighted them in the revised manuscript. We have checked the written language to make the expression clearer.

Question 5. In page 10, the authors described “4T1 tumour cells were treated with free OLE, PEG-T^{ECM}-NS/OLE and PEG-T^{ECM}-NS/OLE pretreated at pH 6.8 with 10 mM H₂O₂ for 12 h. Why this experiment use 10 mM H₂O₂ needs explanation. Because the authors used 100 μM H₂O₂ in all other experiments.

Reply: In page 10, the description of 10 mM H₂O₂ was spelling mistake. In this experiment, we also used 100 μM H₂O₂. We have carefully checked the written language in the manuscript.

Question 6. In figure 8a, the authors should confirm the contrast is consistent in all imaging. The scale intensity bar should be added.

Reply: The figure 8a mentioned in the question may be figure 5a. We have added the scale intensity bar in the revised manuscript.

Question 7. In figure 8, it would be interesting to test if the ROS within the tumor was reduced after treatment.

Reply: As shown in Figure R3, we have conducted additional experiment to test if the ROS within the tumor was reduced after treatment. We began our study by establishing orthotopic colorectal cancer model with CT26 tumor cells. The orthotopic implantation methodology has been described in detail in the literature [*Adv. Mater.* 2018, 1805007, *Nat. Biomed. Eng.* 2019, 3, 717-728]. Intraperitoneal injection of L-012 into orthotopic colorectal cancer mice led to the detection of ROS scavenging by nanoparticles at the tumor site. PEG-T^{ECM}-NS treatment had reduced bioluminescence signal compared to PEG-T^{ECM}-NS without ROS scavenging ability (Figure R3a). Quantification of bioluminescence signals indicated the ROS level in the group of PEG-T^{ECM}-NS without ROS scavenging ability was 3.41 and 2.52-fold higher than those of PEG-T^{ECM}-NS at 24 h and 72 h, respectively (Figure R3b). We investigated the tumor inhibition of PEG-T^{ECM}-NS/OLE in orthotopic CT26 tumor mice. As shown in Figure R3c, PEG-T^{ECM}-NS/OLE showed 91.9% reduction in the colon tumor number. PEG-T^{ECM}-C₁₈/OLE without ROS scavenging ability achieved 68.8% reduction. As shown in Figure R3d, no significant change in the body weight of mice treated with PEG-T^{ECM}-NS/OLE was observed during treatment, verifying the biocompatibility and safety of PEG-T^{ECM}-NS/OLE. We performed immunofluorescence analysis to study the expression of HMGB1, CRT, and CD8⁺ T cells in the tumour tissues after different treatments. We observed the detectable increase of HMGB1 and CRT in the tumor from the group of PEG-T^{ECM}-NS/OLE, demonstrating that the ICD was efficiently elicited (Figure R3e). PEG-T^{ECM}-NS/OLE treatment increased CD8⁺ T cell permeation as compared to PEG-T^{ECM}-C₁₈/OLE. The expression of HMGB1 and CRT protein from orthotopic CT26 tumor treated with the PEG-T^{ECM}-NS/OLE was obviously elevated, which showed a similar induction trend with immunofluorescence analysis (Figure R3f). Caspase-3 and H&E staining of the tumour tissue

of mice treated with PEG-T^{ECM}-NS/OLE showed extensive tumour cell death (Figure R3g). The levels of cytokine secretion in serum were measured by ELISA to monitor the nanoscavenger induced immune response. PEG-T^{ECM}-NS/OLE group displayed a significantly increase in IL-12p40, IFN- γ , and TNF- α expressions in the serum, which confirmed that nanoscavenger effectively induced ICD and improved the immune responses. The proportion of CD8⁺ T cells in the tumors of mice with treatment of PEG-T^{ECM}-NS/OLE showed obviously increase compared to PEG-T^{ECM}-C₁₈/OLE without ROS scavenging ability (Figure R3i). Furthermore, PEG-T^{ECM}-NS/OLE induced a higher proportion of IFN- γ +CD8⁺ T cells in spleens than PEG-T^{ECM}-C₁₈/OLE (Figure R3j and k). t-SNE analysis the tumor tissues from mice after different treatments indicated that increased INF- γ ⁺CD8⁺T cells, CD8⁺ T cells and T memory cells infiltration into tumors treated PEG-T^{ECM}-NS/OLE.

Figure R3 In vivo antitumour and ROS scavenging activity of PEG-T^{ECM}-NS/OLE in orthotopic colorectal cancer models with CT26 tumor cells. (a) In vivo chemiluminescence imaging to measure the ROS in tumor microenvironment by L-012. (b) Quantification of L-012 chemiluminescence. (c) Colon tumor numbers after different treatment. (d) Body weight curves of orthotopic colorectal cancer mice after different treatments. (e) Immunofluorescence of HMGB1, CRT, and CD8⁺ T cells in the tumour tissues after different treatments. (f) Western blot analysis of HMGB1 and CRT expression levels in the tumour tissues. (g) Caspase-3 staining and H&E staining of the tumour tissues with different treatments. (h) Quantification of secretion of IL-12p40, IFN- γ and TNF- α in sera from mice. (i) CD8⁺ T cells in 4T1 tumour analyzed by flow cytometry. (j) The determination of IFN- γ positive CD8⁺ T cells (CD8⁺IFN- γ ⁺ T cells) within tumour by flow cytometry. (k) Quantitative analysis of IFN- γ ⁺CD8⁺ T cells. (l) T-distributed stochastic neighbor embedding (t-SNE) visualization of clustering of representative markers of cells from tumour detected by flow cytometry, each dot corresponds to one single cell.

Question 8. Please clearly state sample size and how many times are repeated for each experiment. Please add the statistical information in many figures. Please add the scale bar in figures (e.g. fig 5k).

Reply: We have clearly stated sample size and the repeated times in the revised manuscript. We have added the statistical information and scale bar in figures.

Reviewer #2 (Remarks to the Author): Expert in ROS scavenging and nanotherapy

In the manuscript, the authors prepared a tumor extracellular matrix (ECM) targeting ROS nanoscavenger. Dual-benzaldehyde terminated polyethylene glycol as a caging polymer was introduced to construct the cross-linked “stealth” delivery system (PEG-TECM-NS) with pH sensitive imine bonds. This intelligent nanoscavenger that can sweep away the ROS from tumor microenvironment to relieve the immunosuppressive ICD elicited by specific chemotherapy based on oleandrin (OLE) anticancer drug and prolong the survival of T cells for “personalized” cancer immunotherapy. When arriving at the tumor site, the de-shielding of PEG corona triggered by tumor acidity leads to the exposure of ECM targeting peptide and anchor on the ECM, allowing TECM-NS to continuously scavenge extracellular ROS. Meantime, TECM-NS was oxidized by ROS inducing the disruption to release OLE. Free OLE was internalized into tumor cells and induced ICD through the release of HMGB1 from the dying cells. Therefore, the nanoscavenger anchored on the ECM to sweep away the ROS from tumor microenvironment to relieve the immunosuppressive ICD elicited by specific chemotherapy and prolong the survival of T cells for personalized cancer immunotherapy. The nanoscavenger presented by authors is interesting, and the manuscript is well written. This work appears suitable for publication in Nature Communications after minor revision.

Reply: Thanks a lot for your constructive comments and suggestions. We have made changes accordingly.

Question 1. In the manuscript, the authors said that “Recent studies revealed that the immunogenic cell death (ICD) elicited by specific chemotherapy or radiotherapy makes their corpses ‘visible’ to dendritic cells (DCs) that present antigens to T cells with specific antitumor immune responses, which then control residual tumor cells” Could you give the detail examples of the ICD elicited by chemotherapy.

Reply: As a systemic agent, chemotherapy has the potential to initiate an immune response in multiple sites. Doxorubicin (DOX) is a *bona fide* ICD inducer that has already been widely evaluated [Angew. Chem. 2018, 130, 11938-11943, Biomaterials 2020, 230, 119659, Nat. Commun. 2017, 8, 1811]. The antitumor efficacy and immunity induced by DOX can be enhanced by combination with immunotherapy. Paclitaxel (PTX), is also known to induce ICD [Adv. Funct. Mater. 2020, 30, 1906605, Adv. Healthcare Mater. 2020, 9, 1901858]. Furthermore, nanoparticles mediated chemotherapy has been reported to enhance ICD and consequently improve antitumor effects of the free ICD inducer. Oxaliplatin encapsulated in nanoparticles released more DAMPs and induced more dendritic cell and T lymphocyte activation and infiltration than free oxaliplatin, improving anticancer efficacy in immunocompetent mice [Biomaterials 2016, 102, 187-197, Adv. Mater. 2018, 30, 1803001, Adv. Mater. 2020, 2002160].

Question 2. The in vitro accumulated drug release was very important. The results of drug release can confirm the stability of cross-linked nanoparticles and the sensitive drug release. The release rate of OLE could be significantly enhanced in pH 6.8 with 10 mM H₂O₂. To show the data more clearly, the authors should give the detail data of cumulative drug release in Figure 2g.

Reply: The accumulated drug release curve of PEG-T^{ECM}-NS at pH 7.4 and 6.8 with or without 10 mM H₂O₂ was shown in Fig. 2g. The drug release rate of PEG-T^{ECM}-NS at pH 7.4 with or without 10 mM H₂O₂ was negligible and less than 25% in 24h, indicating that the crosslinked structure of PEG-T^{ECM}-NS was relatively stable under neutral condition. The accumulative amount of OLE from PEG-T^{ECM}-NS at pH 6.8 with the presence of 10 mM H₂O₂ was 3.21-fold compared with that at pH 7.4 with or without 10 mM H₂O₂ in 24 h. The release rate of OLE was significantly enhanced at pH 6.8 with the presence of 10 mM H₂O₂, under which the PEG coating on the surface of T^{ECM}-NS was de-shielded due to the breakage of pH sensitive imine bonds. Despite the removal of PEG, T^{ECM}-NS was able to maintain the intact nanoparticle structure (Supplementary Figure 16 and 17). Therefore, without H₂O₂, the drug release rate of PEG-T^{ECM}-NS at pH 6.8 was only about 28% and slightly faster than under physiological condition (about 23%). The significantly increased OLE release from PEG-TECM-NS at pH 6.8 with 10 mM H₂O₂ (Fig. 2g) was attributed to pH-induced de-shielding of PEG and hydrophilic transition of thioether to sulfoxide, inducing the disassembly of the nanoparticles. Through the oxidation of thioether residues in PPS segments to sulfoxides, PEG-T^{ECM}-NS have the capacity to scavenge ROS (Supplementary Figure 18).

Question 3. The binding affinity of PEG-T^{ECM}-NS and T^{ECM}-NS on collagen was measured by enzyme-linked immunosorbent assay (ELISA) using biotinylated nanoparticle and substrates coated with rat tail collagen type I. Please provide the data of 4T1 tumors pieces treated with free collagen targeting peptides.

Reply: We have added 4T1 tumour data as suggested (Figure R4).

Figure R4 CLSM images of the slices sectioned from the 4T1 tumour pieces (1 cm³) treated with free collagen targeting peptides (T^{ECM}) for about 4 h. Blue channel, nucleus; green channel, collagen and red channel, RB-labelled T^{ECM}.

Question 4. Free OLE was internalized into tumor cells and induced ICD. ICD of tumor cells is characterized by inducing extracellular release of HMGB1 as “find me signals” and cell surface expression of calreticulin (CRT) as “eat me signals”. The release of ATP was also a characteristic of ICD. Extracellular secreted and intracellular distributed ATP from tumor cells was measured as shown in Fig. 3f. Please explain the difference between the groups.

Reply: have added detailed explanation of the extracellularly secreted and intracellularly distributed ATP from tumour cells. ATP secretion was evaluated by ATP assay to further verify the ICD induction property of PEG-T^{ECM}-NS/OLE. We found the intracellular ATP in the control group was significantly higher than that of PEG-T^{ECM}-NS/OLE pretreated at pH 6.8 with 100 μM H₂O₂ group. The ATP secretion in the cell culture medium of PEG-T^{ECM}-NS/OLE pretreated at pH 6.8 with 100 μM H₂O₂ group was 6.93 fold higher than that of control group (Figure 3f). These results suggest that ICD was elicited by OLE and PEG-T^{ECM}-NS/OLE pretreated at pH 6.8 with 100 μM H₂O₂.

Question 5. To evaluate the efficacy of PEG-T^{ECM}-NS/OLE induced immunogenicity of the tumor cells and turned the tumor cells into antigen-presenting cell (APC) via ICD, the cell surface specific expression of CD80 and CD86 marker of bone marrow dendritic cells (BMDCs) separated from BALB/c mice maturation induced by ICD was investigated using flow cytometry. Please provide the detail experiment process.

Reply: have added the detailed experiment process in the revised manuscript.

Question 6. Splenocytes were isolated and stimulated with concanavalin A (ConA) and then incubated with or without 100 μM H_2O_2 or PEG-T^{ECM}-NS. T-cell proliferation was measured by carboxyfluorescein (CFSE) dilution. What's the function of ConA. Please give some literatures.

Reply: Concanavalin A (ConA) is an antigen-independent mitogen and functions as signal one inducer, leading T cells to polyclonal proliferation [J. Immunol. 1981, 126, 1185-1191, Proc. Natl. Acad. Sci. U S A. 1983, 80, 3466-3469].

Question 7. In the manuscript, the authors also examined the inhibition of T cell proliferation under different concentration of H_2O_2 in vitro. Please provide the quantitative results.

Reply: We have added the quantification results (**Figure R5**) in the revised supporting information.

Figure R5 Quantification of T cell proliferation percentage.

Question 8. Could you give the quantitative analysis of the MRI results in Fig. 5e.

Reply: We have added the quantitative analysis of the MRI results in the revised supporting information (Figure R6).

Figure R6 Semi-quantitative analysis of the signal-to-noise ratio (SNR) in tumors.

REVIEWERS' COMMENTS

Reviewer #1 (Remarks to the Author):

The authors have adequately addressed my prior comments.

Reviewer #2 (Remarks to the Author):

The work has been well revised. The present version seems to be acceptable for publication.